# Visual Function in Alzheimer’s Disease: Current Understanding and Potential Mechanisms Behind Visual Impairment

**DOI:** 10.3390/jcm14175963

**Published:** 2025-08-23

**Authors:** Tania Alvite-Piñeiro, Maite López-López, Uxía Regueiro, Juan Manuel Pías-Peleteiro, Tomás Sobrino, Isabel Lema

**Affiliations:** 1Corneal Neurodegeneration Group (RENOIR), Clinical Neurosciences Research Laboratory (LINC), Health Research Institute of Santiago de Compostela (IDIS), 15706 Santiago de Compostela, Spain; tania.alvite.pineiro@rai.usc.es (T.A.-P.); maite.lopez.lopez@rai.usc.es (M.L.-L.); uxia.regueiro.lorenzo@usc.es (U.R.); 2Department of Surgery and Medical-Surgical Specialties, Faculty of Optics and Optometry, Universidade de Santiago de Compostela (USC), 15706 Santiago de Compostela, Spain; 3NeuroAging Group (NEURAL), Clinical Neurosciences Research Laboratory (LINC), Health Research Institute of Santiago de Compostela (IDIS), 15706 Santiago de Compostela, Spain; tomas.sobrino.moreiras@sergas.es; 4Center for Networked Biomedical Research on Neurodegenerative Diseases (CIBERNED), Carlos III Institute of Health, 28029 Madrid, Spain; 5Department of Neurology, Clinical University Hospital of Santiago de Compostela, 15706 Santiago de Compostela, Spain; 6Instituto Galego de Oftalmoloxía (INGO), Provincial Hospital of Conxo, 15706 Santiago de Compostela, Spain

**Keywords:** Alzheimer’s disease, brain, color vision, contrast sensitivity, ocular motility, perception, retina, visual acuity, visual field, visual pathways

## Abstract

Alzheimer’s disease (AD) is the leading cause of dementia worldwide and is becoming one of the most morbid diseases of this century. Recently, ocular research in AD has gained significance, as the eye, due to its close relationship with the brain, can reflect the presence of neurological disorders. Several studies have reported alterations in various ocular structures in AD, ranging from tear fluid to the retina. These changes, particularly in the retina and the optic nerve, along with cerebral atrophy affecting visual brain areas, may lead to visual dysfunctions. This narrative review summarizes and critically examines current evidence on these impairments and explores their possible underlying mechanisms. A decrease in visual acuity, contrast sensitivity, and color vision has been observed, primarily associated with retinal ganglion cell loss or damage. Furthermore, alterations in the visual field, ocular motility, and visual perception have been recorded, mainly resulting from cortical changes. These optical parameters frequently correlate with patients’ cognitive status. In conclusion, these findings highlight the importance of developing strategies to preserve visual function in these patients, helping to prevent further deterioration in their quality of life, and emphasize the potential of visual function assessment as a tool for diagnosis or predicting AD progression.

## 1. Introduction

Dementia is defined as an acquired syndrome that causes cognitive impairment, interfering with the affected individual’s work and domestic and social life [1]. Its hallmark symptoms include memory deficits, language difficulties, impaired problem-solving abilities, and other cognitive impairments that limit an individual’s capacity to perform activities of daily living [2]. Dementia can be caused by neurodegenerative diseases, such as Alzheimer’s disease (AD) or Lewy body dementia, or by non-neurodegenerative conditions, including brain trauma, brain tumor, or vascular dementia [1].

Currently, dementia affects approximately 50 million people worldwide, and this number is expected to triple in 2050 [3]. The incidence of dementia increases with age. Among individuals aged 65 and older, the incidence ranges from 5–8%, but this value rises to 20–25% for those aged 85 and above [4]. Additionally, dementia affects a higher number of women than men, which may be attributed to both genetic factors and the longer life expectancy of women [5].

AD is the leading cause of dementia, accounting for 60–80% of diagnoses [2]. It is a progressive neurodegenerative disorder primarily characterized by an abnormal accumulation of proteins in the brain. Additionally, other pathological mechanisms such as neuroinflammation, oxidative stress, and synaptic dysfunction occur. All these processes result in the loss of neuronal connections, neuronal death, and brain tissue loss, leading to a cognitive decline that produces the loss of patient independence [4].

Specifically, AD is becoming one of the most severe and morbid diseases of this century [6]. It is associated with significant socioeconomic burdens on the healthcare sector, as well as financial and public health challenges for society as a whole [7]. Even though AD has been recognized for over 100 years, the molecular mechanisms that trigger and drive disease progression remain unclear [8]. However, various macro and microscopic markers associated with the pathology are known, which aid in its characterization [9]. At the macroscopic level, hippocampal and cortical atrophy is observed due to neuron loss in these areas, which is accompanied by a decline in cognitive functions (Figure 1). The degree of atrophy may worsen as the patient ages and the disease progresses [10].

At the microscopic level, the formation of extracellular senile plaques composed of beta-amyloid peptides (Aβ) and the accumulation of hyperphosphorylated tau protein, which forms neurofibrillary tangles (NFT) in the cytoplasm of neurons, are prominent features (Figure 1) [11]. The Aβ peptide is generated through cleavage by secretases (α, β, γ) from the amyloid precursor protein (APP), a protein involved in brain development, memory, and synaptic plasticity [10,12]. Under physiological conditions, Aβ has a protective role in synapses [11]. However, due to genetic mutations or variations in the concentration of certain neuropeptides, an imbalance between the synthesis and clearance of Aβ can occur [4]. This imbalance disrupts metabolism, leading to the aggregation and extracellular accumulation of these proteins, resulting in the formation of senile plaques (mainly composed of Aβ peptides of 40 and 42 amino acids) [12,13]. Elevated concentrations of Aβ in the brain lead to neuronal toxicity and degeneration [11].

NFTs are abnormal filaments of hyperphosphorylated tau protein that pair and helically coil in the cytoplasm of neurons [14]. Tau protein is a neuronal protein that is associated with microtubules, helping to maintain cytoskeleton stability. The binding to microtubules is mediated by the phosphorylation of serine/threonine residues of the proteins, carried out by kinases. The abnormal hyperphosphorylation of tau, resulting from genetic mutations or kinase dysregulation, leads to a loss of its affinity for microtubules and the formation of NFT. Consequently, tau loses its ability to maintain neuronal structure [4]. Furthermore, the deposition of NFT disrupts communication between neurons and signal processing, ultimately leading to cellular apoptosis [15].

Other mechanisms also play an important role in the pathogenesis of AD. First, it is known that there is an activation of microglia and an inflammatory response, both of which contribute to neurotoxicity [15]. Additionally, alterations related to oxidative stress have been observed in patients with AD [9,10]. The brain is an organ with an extraordinarily high metabolism. Under resting conditions, brain metabolism is about 7.5 times the average metabolism of the rest of the body [16]. This makes it particularly susceptible to reactive oxygen species (ROS). When ROS production becomes excessive, they can interact with neurons, leading to lipid oxidation or mitochondrial dysfunction, which results in neuronal cell death [9,10].

Although most AD cases are sporadic, meaning they lack a dominant genetic cause, certain mutations have been identified that can increase the risk of developing the disease [9]. Specifically, early-onset AD (which manifests in individuals under 65 years of age) is associated with mutations in the *APP* (*amyloid precursor protein*), *PSEN1* (*presenilin 1*) and *PSEN2* (*presenilin 2*) genes. Late-onset AD (which occurs in individuals over 65 years of age) is linked to the presence of the ε4 allele in the *APOE* (*Apolipoprotein E*) gene [11].

Early diagnosis of AD is essential for the timely administration of effective treatments [17]. Current medications can alleviate symptoms and slow the progression of cognitive decline, so their early-stage administration could optimize these benefits [18]. Recently, a new treatment called Lecanemab gained approval from both the Food and Drug Administration (FDA) and the European Medicines Agency (EMA) [19,20]. This drug, a humanized IgG1 monoclonal antibody that specifically binds to Aβ protofibrils, demonstrated in phase III clinical trials that it reduced amyloid markers in early AD and resulted in less cognitive decline and less increase in caregiver burden than patients receiving a placebo at 18 months [21,22].

However, pathological changes in AD begin up to two decades before the appearance of the first symptoms. Therefore, diagnosing AD at the onset of symptoms is not very useful, because cognitive decline may already be advanced, with irreversible neuronal damage [4,17]. For this reason, the identification of biomarkers, particularly in the early stages of AD, is crucial, as they can enhance diagnostic accuracy, increase the understanding of disease mechanisms, and provide valuable information for the development of new therapeutic targets. An ideal biomarker would enable the identification of the pathology before the onset of symptoms, would be easily accessible for mass screening in at-risk populations, and would allow for the exclusion of other related diseases [23].

In recent years, the study of the eye has emerged as a promising alternative for the identification of biomarkers, as it is an extension of the central nervous system (CNS) and thus can reflect the presence of various neurological disorders. Additionally, the study of the eye allows for minimally invasive assessments [24,25]. Previous studies have reported the presence of ocular changes in patients with AD. Alterations have been observed in tear fluid, cornea, aqueous humor, pupil, vitreous humor, choroid, retina, and optic nerve [17,25]. These changes, particularly within the retina and optic nerve, along with the involvement of the visual cortex, may lead to visual function alteration in AD patients [26].

This review compiles and critically examines current knowledge regarding the visual function (visual acuity, contrast sensitivity, color vision, visual field, ocular motility, visual perception, and stereopsis) in individuals with AD, as well as the potential underlying causes of their visual dysfunction. Its objective is to characterize the visual profile typically associated with the condition. Furthermore, by consolidating existing evidence, this article explores the potential of visual impairments to serve as accessible clinical markers for early diagnosis or for monitoring disease progression through cost-effective and minimally invasive tests.

## 2. Visual Information Processing

Visual information processing begins in the retina, which is responsible for converting light into electrical impulses through photoreceptors (rods and cones). These electrical impulses stimulate the retinal ganglion cells (RGCs) via the bipolar cells, while horizontal and amacrine cells modulate this transmission. The axons of the RGCs converge to form the optic nerve, through which information is transmitted to the brain (Figure 2a) [27]. Upon reaching the optic chiasm, the nerve fibers from the nasal retina decussate, causing each visual hemifield to be represented in the contralateral visual areas of the brain [28]. After decussation, the fibers from the nasal retina of one eye, along with those from the temporal retina of the other eye, proceed along the optic tract to the lateral geniculate nucleus (LGN) (Figure 2b) or other extrageniculate visual structures [27].

Most nerve fibers reach the LGN, located in the thalamus, from where axons are sent to the primary visual cortex (V1) [28]. Additionally, another geniculated pathway forms with subcortical connections to the quadrigeminal tubercles, which are involved in the regulation of ocular motility, and a third pathway extends to extracalcarine associative visual areas [27]. The LGN has a laminar structure composed of 6 different layers. Layers 1 and 2 give rise to the magnocellular pathway (M pathway), which specializes in processing depth perception and motion. Layers 3 to 6 give rise to the parvocellular pathway (P pathway), which is related to shape and color perception. Nerve fibers from the temporal retina of the ipsilateral eye project their axons onto layers 2, 3, and 5, while nerve fibers from the nasal retina of the contralateral eye project onto layers 1, 4, and 6 of the LGN. Thus, each layer receives information from a single eye [28]. Moreover, a third pathway, known as the koniocellular pathway, connects to diffuse layers of the LGN located between the main layers. This pathway is involved in color vision [29].

On the other hand, a small percentage of nerve fibers follow the extrageniculate pathway. Most of these fibers project to the superior colliculus, while the remainder reach other structures of the CNS. The superior colliculus is organized into layers. The more superficial layers detect the movement of light stimuli, while the deeper layers are involved in the coordination of ocular, head, and body movements [28].

From the primary visual cortex, cells send their axons to extrastriate visual areas such as V2, V3, V4, and V5 [28]. At this point, two specialized streams can be differentiated: the ventral stream and the dorsal stream. The ventral stream, or the “what” pathway, projects to the inferior temporal lobe and is involved in visual object recognition. In contrast, the dorsal stream, or the “where” pathway, projects to the posterior parietal cortex and is responsible for processing spatial information [30].

In AD, macroscopic atrophy of the cerebral cortex occurs, typically with a predominance in the temporal cortex and the associative frontal and parietal areas [31]. Specifically, regarding the visual pathway, it has been reported that AD patients exhibit a loss of RGCs, predominantly affecting the cells of the magnocellular pathway [32]. Furthermore, it has also been suggested that this disease affects the dorsal visual pathway [33]. Subsequent sections explore the relationship between defects in visual function and their potential cortical or retinal causes.

## 3. Visual Acuity

Even though some authors found no significant differences in visual acuity (VA) between AD patients and healthy control subjects [34,35], other studies have reported a loss of VA in these patients [36,37,38] (Table 1). Salobrar-García et al. [36] observed a statistically significant decrease in VA, measured using the Snellen test, in ophthalmologically healthy subjects with mild and moderate AD compared to the age-matched control group. Elvira-Hurtado et al. [37] also found a reduction in VA in patients with mild cognitive impairment (MCI), mild AD, and moderate AD compared to control subjects. In contrast, Nolan et al. [38] found that the best-corrected VA, assessed using the logMAR Early Treatment Diabetic Retinopathy Study (ETDRS) chart at 6 m, was significantly lower in AD patients than in the control group. However, when adjusting the results for patient age, macular pigment, diet, education, and the presence of age-related macular degeneration (AMD), the result was no longer statistically significant. Despite a higher prevalence of AMD in AD patients, the data suggest that their visual loss is not associated with this condition, as AD patients without AMD also exhibited a reduction in VA [38].

Some studies have assessed the correlation between scores on the Mini-Mental State Examination (MMSE) and the VA levels in patients with AD. A statistically significant direct correlation between the two variables was observed. These findings indicate that a decline in VA is associated with a reduction in cognitive capacity in affected individuals [36,37].

The heterogeneity of the results observed may be primarily attributed to methodological differences across studies, including the type of VA test used (such as Sloan letters, logMAR ETDRS, and Snellen chart), as well as variations in the AD stages examined. Some studies assess disease severity using the Clinical Dementia Rating (CDR), while others rely on the MMSE, which may lead to inconsistencies by classifying patients with similar clinical characteristics into different severity groups. Given that the studies by Rizzo et al. [34] and Polo et al. [35] focused on patients with mild AD and reported no significant differences, it is possible that VA impairment becomes more evident in individuals with more advanced stages of AD.

Visual loss associated with AD may be attributed to the degeneration of the optic nerve and retina that occurs during the progression of the disease [39]. Retinal pathological changes observed in AD patients closely mirror those occurring in the brain, including Aβ accumulation, apoptosis, inflammation, and oxidative stress. These processes disrupt the visual pathway and are believed to contribute to the visual dysfunction observed in AD, including reduced VA. The brain cells that mediate these neurodegenerative processes in AD are microglia, astrocytes, and neurons (Figure 3) [40]. Microglia play a dual role in retinal degeneration in AD. In early stages, they exert neuroprotective functions by promoting Aβ clearance and releasing anti-inflammatory cytokines. However, with prolonged activation, they shift to a proinflammatory state, characterized by excessive cytokine production, increased oxidative stress, and impaired Aβ clearance. This process contributes to RGC loss, synaptic dysfunction, and thinning of the retinal nerve fiber layer (RNFL). Moreover, hyperactivation of microglia can disrupt vascular homeostasis, thereby reducing retinal perfusion and intensifying hypoxic conditions [40,41]. Astrocytes, under chronic pathological conditions, undergo reactive gliosis and release proinflammatory mediators that further exacerbate retinal damage. Furthermore, alterations in astrocyte angiogenic activity can lead to abnormalities in the retinal vasculature [40,42]. Lastly, many studies have reported a significant reduction in RGCs and RNFL thickness in AD patients that can be a consequence of a retrograde degeneration of the optic nerve and retinal layers, which is caused by the cerebral pathology of AD [40].

In 2017, Polo et al. [35] published the results of a study in which they assessed VA at three different contrast levels (ETDRS test) in individuals with AD and correlated it with structural changes in the retina of these patients. Initially, they observed a reduction in the best-corrected VA in AD patients compared to control subjects across all contrast levels (100%, 2.50%, 1.25%), although no significant differences were found. Nevertheless, ETDRS VA was associated with macular thickness and the thickness of the RNFL. Specifically, VA at 100% contrast was significantly correlated with the thickness of the outer macular sectors and with the average, superior, and inferior thickness of the RNFL. It is important to note that while VA primarily depends on the fovea, in AD patients, visual loss may result from a generalized loss of RGCs, resulting from pathological changes in the retina associated with the disease [35].

## 4. Contrast Sensitivity

Contrast sensitivity (CS) is the ability of the visual system to distinguish between an object and the background on which it is located [38]. Generally, it has been observed that patients with AD exhibit a reduction in CS across all spatial frequencies [36,38]. However, some studies report a greater reduction in high spatial frequencies [37,43], while others indicate that the decrease is more pronounced in low spatial frequencies [35]. Additionally, some studies have found no significant differences in CS between AD patients and the control group [44] (Table 2).

Salobrar-García et al. [36] assessed the level of CS in patients with mild and moderate AD compared to healthy age-matched control subjects using the CSV-1000E test (VectorVision, Greenville, OH, EE. UU). A statistically significant decrease in CS was observed across all evaluated spatial frequencies (3, 6, 12, and 18 cpd) in AD patients compared to the control group. Specifically, the spatial frequency of 18 cpd was the most affected. An analysis with area under the receiver operator characteristic (aROC) curves revealed that the test with the highest predictive value was CS at high spatial frequencies [36]. On the other hand, Polo et al. [35] evaluated CS in AD patients compared to healthy subjects using the Pelli-Robson test and the CSV-1000E test. Both tests showed a significant decrease in CS in AD patients. However, in this case, the CSV-1000E test, which measures different spatial frequencies (3, 6, 12, and 18 cpd), did not show significant differences between the two groups at the 18 cpd frequency [35].

The diversity in the results obtained may be attributed to differences between the subjects included, as each study incorporates patients at various stages of the disease, classified according to different criteria, and the different tests employed to assess CS [36]. Neargarder et al. [45] conducted a study in which they compared CS, measured using different tests, in a group of AD patients. They found that VA had a greater impact on some of the CS measures. Specifically, the Regan Low Contrast Letter Acuity and Vistech VCTS 6500 tests were influenced by differences in VA, while the Pelli-Robson and Freiburg Visual Acuity Test were not affected by this variable [45].

In addition, small sample sizes in certain studies, particularly those by Hutton et al. [43] and Schlotterer et al. [44], may have limited statistical power, reducing the reliability and consistency of their findings.

The CSV-1000E test offers certain advantages over the LogMAR ETDRS and Pelli-Robson tests, particularly in cognitively impaired populations. Notably, it does not require verbal responses from participants, which can increase reliability in individuals with language and memory difficulties. When focusing on studies that used the CSV-1000E test [35,36,37], Salobrar-García et al. [36] and Elvira-Hurtado et al. [37] reported significant differences in CS, especially at high spatial frequencies. However, Polo et al. [35] did not observe such differences at 18 cpd. This discrepancy might be due to differences in how disease severity was classified. In the study by Polo et al. [35], patients with mild and moderate AD were grouped together, with disease severity assessed using MMSE scores. Their sample included participants with a mean MMSE of 15.54 ± 7.1, indicating a broader range of cognitive impairment. In contrast, Salobrar-García et al. [36] and Elvira-Hurtado et al. [37] only incorporated patients with an MMSE score above 17. The inclusion of individuals with lower neuropsychological test scores may have led to less reliable results, as patients with more advanced stages of the disease are often less cooperative and have greater difficulty completing visual tasks accurately.

Moreover, focusing specifically on the study by Elvira-Hurtado et al. [37], which is the only one to evaluate patients with MCI, it was observed that these patients showed significant differences from control subjects only at spatial frequencies of 12 and 18 cpd. This finding may suggest that CS loss in AD patients begins at higher spatial frequencies and gradually extends to lower spatial frequencies as the disease progresses.

Polo et al. [35] investigated the association between CS and retinal abnormalities in patients with AD. They observed that CS, measured by the Pelli-Robson test, was significantly correlated with macular thickness and the thickness of the RNFL in most of the evaluated sectors. On the other hand, measurements obtained using the CSV-1000E were closely associated with the majority of macular parameters as well as the average and superior region thickness of the RNFL [35].

Finally, several studies have assessed whether there is a correlation between the score obtained on the MMSE and the measurement of CS in patients with AD. A statistically significant direct correlation was found between the cognitive test scores and CS measurement across all spatial frequencies evaluated [36,37]. Salobrar-García et al. [36], despite recording this correlation between the two variables, observed no significant changes in CS between patients with mild and moderate AD. These results suggest that CS deterioration is more pronounced in the early stages of the disease, followed by stabilization as the pathology progresses. As such, CS assessment may hold potential as a clinical marker for the early evaluation of AD, particularly at high spatial frequencies [36].

In the retina, two main types of ganglion cells can be distinguished: parvocellular and magnocellular cells, which give rise to the P and M pathways, respectively. The P pathway is composed of smaller and more numerous cells and is more sensitive to high spatial frequencies. In contrast, the M pathway consists of larger and less numerous cells and is more sensitive to low spatial frequencies [46]. Some studies suggest that the loss of CS observed in patients with AD is due to a generalized loss of both cell types [35]. It has been proposed that this alteration in ganglion cells, specifically in the RNFL, may occur in the early stages of AD, even before hippocampal damage, which affects memory, becomes evident [47].

Moreover, CS has shown a correlation with cerebral Aβ and tau deposition, both in patients with established dementia and in individuals at high risk of developing AD, such as those with subjective cognitive decline or MCI. This suggests that CS may serve as a useful non-invasive biomarker for detecting AD-related pathophysiological changes [48].

## 5. Color Vision

Color vision is based on three primary axes: the protan axis (associated with red perception), the deutan axis (associated with green perception), and the tritan axis (associated with blue perception). Each of these axes is linked to a specific type of cone in the retina (L, M, and S cones, respectively) [49]. There is some controversy regarding the influence of AD on color perception. While some studies have found no significant differences in color vision between AD patients and control subjects [50,51], others have reported various alterations [35,36,37,49,52,53] (Table 3).

Pache et al. [49] assessed color vision in patients with AD compared to a control group using two tests: the PV-16, which detects acquired deficits along the protan, deutan, and tritan axes, and the Ishihara test, which identifies deutan and protan abnormalities. Both assessments revealed that AD patients made more nonspecific errors than control subjects. However, no correlation was found between the results of the Ishihara and PV-16 tests and the severity of the disease [49].

Salobrar-García et al. [52] used the Roth 28-Hue test to assess color perception in patients with mild AD compared to age-matched healthy controls. This test did not reveal any significant differences in overall color vision between groups. However, a more detailed analysis showed a significant increase in nonspecific tritan errors among AD patients. In a subsequent study, Salobrar-García et al. [36] applied the same test to subjects with mild and moderate AD compared to control subjects. In this case, an analysis of the total nonspecific errors revealed that both AD groups made significantly more errors than the control group. Additionally, independent analysis of the deutan and tritan axes also showed a greater number of errors in the study groups (mild and moderate AD) for both axes. Furthermore, a significant inverse correlation was found between MMSE scores and the total number of errors, as well as with errors in the deutan and tritan axes [36].

Vidal et al. [53] observed a significant decrease in color vision in patients with AD compared to age-matched control subjects for all three axes. Additionally, worse color perception was noted in subjects with MCI on the tritan axis. This assessment was conducted using the Cambridge color test, a computerized psychophysical test based on the same principle as the Ishihara plates. Moreover, Vidal et al. [53] also classified individuals according to the A/T/N framework and found that overall color vision loss was present in individuals exhibiting signs of neurodegeneration (N+), as measured by fluorine-18 fluorodeoxyglucose positron emission tomography (18F-FDG-PET), a method assessing cerebral glucose metabolism. However, this alteration was not observed in patients showing only signs of Aβ deposition (A+), as measured by Pittsburgh compound B positron emission tomography (PIB-PET). These findings suggest a possible link between color vision impairment and brain neurodegeneration not exclusively associated with AD, as the presence of Aβ deposits is considered a necessary criterion for inclusion within the disease continuum [53].

The discrepancy in the observed findings may be partly attributed to the variability of the tests used to assess color vision in these patients, which complicates the direct comparison of the results. These tests differ substantially in their cognitive demands, a critical factor when assessing patients with AD. Specifically, tests such as the Ishihara require substantial involvement of higher-order cognitive functions, including working memory, attention, and numerical recognition, which may compromise result reliability in cognitively impaired populations. In contrast, tests like Roth 28-Hue and Farnsworth rely more heavily on visual perception and less on cognitive processing, making them more suitable for individuals with cognitive decline. It should also be noted that performance on these tests may depend on the patient’s memory, underscoring the need to repeat the instructions throughout the process [52].

Moreover, it is important to highlight that the studies reporting no significant differences in color vision between groups [50,51] involved very small sample sizes, which limits the generalizability of their findings. Small cohorts reduce the ability to detect subtle differences between patients and control subjects.

Another limitation of the studies addressing color vision in AD is the inclusion of participants at different stages of the disease. Notably, only the studies by Elvira-Hurtado et al. [37] and Vidal et al. [53] included patients with MCI. While Elvira-Hurtado et al. [37] found no significant differences in color vision between MCI patients and healthy controls, Vidal et al. [53] reported impaired color discrimination along the tritan axis in the MCI group. These discrepancies may stem from methodological differences in the color vision tests employed or variations in the diagnostic criteria used to define MCI. Given the potential of tritan axis alterations as an early indicator of neurodegeneration, further research involving individuals in preclinical stages of AD is warranted to obtain more consistent results.

Polo et al. [35] investigated whether there was a correlation between color vision, assessed through the Farnsworth and L’Anthony tests, and various retinal parameters. A mild positive association was observed with certain isolated sectors of the RNFL and the macular area. However, these patients did not show a significant reduction in foveal thickness, suggesting that the visual function loss is more likely attributable to a general loss of RGCs, which leads to dysfunction in the visual pathways involved in both CS and color vision [35]. It is important to note that, due to their high metabolic demand, RGCs are particularly vulnerable to neurodegenerative damage, such as that caused by mitochondrial abnormalities, axonal transport failures, oxidative stress, and energy depletion, processes that may arise as a consequence of the pathological retinal changes associated with AD [32].

The increased number of nonspecific errors along the tritan axis, observed in many studies, suggests that, in addition to an alteration of the parvocellular pathway (involved in color perception), there is also disruption of the koniocellular pathway [36]. Koniocellular RGCs receive signals from short-wavelength cones (S cones), which are sensitive to the blue-yellow spectrum, and transmit this information to the koniocellular layers of the LGN [52]. Furthermore, some studies have reported a reduction in the area of the cortical V4 region in patients with AD, a region also involved in color vision processing [37,54].

On the other hand, Savaskan et al. [55] observed, in a preliminary study, that in patients with AD, the photoreceptor cell layer was degenerated, and there was reduced expression of the melatonin receptor 1a (MT1), which regulates the specific actions of melatonin. Since melatonin has an antioxidant function, its decrease in these patients is thought to contribute to the degeneration of photoreceptors. This degeneration is not specific to a single type of cone, and therefore, it could impact color vision along any of the primary axes (protan, deutan and tritan) [49].

## 6. Visual Field

Visual field (VF) is defined as the total area perceived by each eye while maintaining a fixed gaze at a particular spot. Under normal conditions, it extends approximately 100° temporally, 60° nasally and superiorly, and 70° inferiorly from the central fixation point. The blind spot, located at the optic nerve head, is found 15° temporally in each eye (Figure 4). VF assessment can be performed using qualitative techniques, such as confrontation campimetry, or quantitative methods, such as computerized perimetry [56]. Computerized perimetry has the advantage of assessing visual sensitivity by presenting light stimuli of varying intensity and small size. In the human visual system, sensitivity is highest at the fovea and decreases towards the periphery. This type of VF measurement enables the detection of dysfunctions at any point along the afferent visual pathway, from the retina to the occipital cortex [57].

The accuracy and reliability of VF tests are closely linked to the patient’s ability to maintain fixation on a specific point and sustain attention to the presented stimuli over time. In addition, subjectivity in the patient’s responses introduces a source of variability that can affect the consistency of the results obtained. Although computer algorithms have been developed to optimize the efficiency of these tests, attentional deficits associated with AD can negatively influence the precision of the results. Furthermore, in older patients, the presence of ophthalmological alterations, such as cataracts or retinal diseases, may alter the VF, highlighting the importance of performing a comprehensive ophthalmological examination before testing to detect any condition that could bias the findings [57].

In 1995, Trick et al. [58] published a study in which they evaluated the VF in patients with senile dementia of Alzheimer’s type compared to healthy control subjects, using computerized perimetry (Humprey 30-2). First, the reliability of the VF results was examined, revealing that patients with dementia had a higher frequency of unreliable VF than healthy controls. These unreliable VF results were excluded from the study to ensure more accurate findings. Subsequently, it was observed that patients with AD exhibited a greater reduction in visual sensitivity than control subjects, with an average defect predominantly affecting the inferior arcuate region but also involving a central alteration. Although arcuate defects are common in other ocular diseases such as glaucoma, in AD, there appears to be a more significant reduction in central sensitivity [58].

Steffes et al. [59] investigated whether patients with AD exhibited greater VF impairment compared to patients with other dementias (secondary dementias due to traumatic brain injury, multiple infarcts, cerebrovascular accidents, or alcoholism). To assess this, they used a hemispheric projection perimetry. A significant reduction in VF was observed in AD patients compared to subjects with other dementias across all evaluated quadrants. Additionally, VF impairment showed a direct correlation with the patient’s cognitive level, showing that greater cognitive deterioration was associated with more pronounced VF alterations [59].

More recently, studies have employed Frequency Doubling Technology (FDT) perimetry to evaluate the VF in patients with AD [60,61,62]. Similarly to previous studies, a higher number of abnormal VF results have been observed in patients with mild AD compared to controls [61]. Additionally, Cesareo et al. [62] found a correlation between the reduction in VF and the increase in cognitive decline. It is worth noting that, compared to traditional perimetry tests, FDT has a shorter test duration and a reduced learning effect [62]. Therefore, due to the facility of performing this test, it may be useful for identifying neurodegenerative diseases associated with visual system alterations, such as AD [60].

Although retinal alterations have been observed in patients with AD [17], it is believed that VF defects may be more closely related to cortical changes, as this disease affects various areas of the visual cortex [58]. Armstrong et al. [63] published a study in which they measured the density of senile plaques and NFT in specific regions of the V1 area to which the superior and inferior visual fields project (lingual gyrus and cuneus, respectively) in patients with AD. The aim of this study was to determine whether the accumulation of these proteins in the cerebral cortex was responsible for the VF defects observed in AD patients. It was found that 39% of patients with AD had a higher accumulation of both proteins in the cuneus, which supports previous findings showing a loss of VF in the inferior arcuate region [63].

Emerging evidence from multimodal studies combining optical coherence tomography (OCT) and magnetic resonance imaging (MRI) has demonstrated that retinal structure alterations in AD patients correlate with a loss of white matter integrity along the visual pathway, decreased volume of the LGN, and reduced functional activity in V1. These findings suggest that retrograde transsynaptic degeneration may occur, whereby neurodegeneration originating in cortical areas propagates backward along the visual pathway, ultimately affecting the LGN and the RGCs [64]. This process may result in reduced visual sensitivity in specific areas of the VF, depending on the region most affected. Therefore, VF defects may serve as a functional indicator of the neurodegeneration along the retino-geniculo-striate pathway. This reinforces its potential role as a non-invasive biomarker for monitoring disease progression. Despite its potential, it is important to recognize that VF testing is a subjective measure that depends heavily on the patient’s ability to respond accurately. Therefore, close supervision during the test is essential, particularly to ensure that the fixation is stable and the patient follows the instructions. Additionally, perimetric reliability indices should be carefully reviewed to identify potentially unreliable results that could bias the findings.

## 7. Ocular Motility

Eye movements are regulated by various brain regions, including the brainstem, basal ganglia, cerebellum, and cerebral cortex [65]. Additionally, ocular motility tests are associated with several cognitive, perceptual, and motor processes, such as attention, working memory, processing speed, motion processing, and inhibition [66]. Given this, the assessment of ocular motility may prove useful in detecting neurodegenerative disorders like AD [65,66]. Ocular movement evaluation tests offer the advantages of being non-invasive, cost-effective, easy to perform, and independent of the patient’s cultural level [66]. Moreover, due to the latest technological advancements, oculomotor alterations can be quantified, enhancing the diagnostic value of these tests [65].

The most studied types of eye movements in patients with AD are fixation, saccades, and smooth pursuit movements [67]. Fixation consists of focusing the vision on a point in space over time and helps to stabilize the image on the retina [65,68]. Saccades are rapid eye movements that allow the fixation to shift from one point to another [65]. Two main types of saccades can be distinguished: prosaccades, which involve shifting gaze towards a target point, and antisaccades, which are movements in the opposite direction to a visual stimulus [65,69]. Lastly, smooth pursuit movements enable following a moving object within the visual field [65].

Several studies have analyzed ocular motility in patients with AD [66,67,68,70,71,72] (Table 4). In 2022, a meta-analysis revealed that patients with MCI and AD exhibit alterations in prosaccadic and antisaccadic latencies, as well as in the frequency of antisaccadic errors. Specifically, antisaccadic movements were shown to be more effective in distinguishing patients from controls [73].

Recently, Qi et al. [66] published a study in which they analyzed lateral fixation, prosaccades, antisaccades, and memory saccades in patients with MCI due to AD, dementia due to AD, and control subjects using the EyeKnow eye-tracking and analysis system. The evaluation of lateral fixation has the advantage of significantly activating the frontal-parietal structures of the ocular movement system. This allows for assessing the function of these brain areas, thereby facilitating the evaluation of neurocognitive functions such as visual attention or inhibitory control in AD patients. In general, ocular movements were found to be impaired in AD patients compared to control subjects. Subsequent analysis revealed that changes in ocular movement parameters were strongly associated with attention, visual memory, and visuospatial ability, suggesting that oculomotor metrics reflect the symptoms of AD patients. Finally, the predictive value of ocular movement parameters for detecting AD was analyzed, and it was observed that the combination of the evaluated parameters was more effective in predicting the progression from control subjects to AD, with an area under the curve of 0.835 and sensibility, specificity, and accuracy of 72.6%, 86.5%, and 76.9%, respectively. On the other hand, the combination of antisaccade accuracy and memory saccades achieved the highest accuracy for predicting the progression from control subjects to MCI due to AD, with an area under the curve of 0.737 and sensibility, specificity, and accuracy of 62.5%, 85.0%, and 71.9%, respectively. This indicates that these parameters are already altered in the early stages of AD, making them promising markers for the early diagnosis of the disease [66].

Tao et al. [67] also investigated eye movements (fixation, saccades, and smooth pursuit movements) in patients with MCI, AD, and control subjects using the EyeKnow eye-tracking system. It was observed that patients with MCI differed from controls in terms of the error correction rate of antisaccades and the total offset degrees of lateral fixation, while AD subjects exhibited a greater number of alterations in fixation and saccadic movements. Smooth pursuit movements did not show significant differences between groups. Notably, many of the oculomotor parameters demonstrated a slight to moderate correlation with cognitive function in the patients, as measured by the MMSE and MoCA tests. Finally, it was found that the combination of lateral fixation evaluation with antisaccades provided the highest predictive value for detecting MCI subjects, with an area under the curve of 0.837 [67].

To assess the relationship between oculomotor parameters and cognitive function, another study was published in 2025 that included patients with mild AD, moderate AD, and control subjects. The analysis showed that metrics related to prosaccadic and antisaccadic tasks were significantly correlated with global cognitive function, as well as with specific cognitive domains (orientation, visuospatial skills, or word fluency). These results highlight the potential of eye movement evaluation as an objective biomarker for the early detection of AD [72]. Additionally, Parra et al. [70] observed that analyzing oculomotor behavior can effectively predict the progression of MCI patients to AD, further demonstrating the potential of this biomarker to assess disease progression.

## 8. Visual Perception and Stereopsis

Visual perception is essential for obtaining information about the world around us, and its disruption leads to a decline in the quality of life for individuals. In patients with AD, visual perception may be impaired due to damage in the associative visual areas of the brain [74].

Specifically, visuospatial perception, which may be impaired in patients with AD, is defined as the ability to identify, integrate, and analyze information related to the position of objects in space. Deficits in visuospatial perception are thought to arise from a dysfunction in the dorsal pathway, which originates in the secondary visual cortex (V2) and projects to the posterior parietal cortex. These impairments are believed to stem from damage to the associative areas where this pathway ends. It is noteworthy that dorsal pathway dysfunction seems to precede alterations in the temporal lobe, which is responsible for episodic memory. As a result, visuospatial processing may serve as a useful biomarker for the early detection of AD [30].

Since stereopsis is a contributing process to visuospatial perception, several studies have analyzed stereoscopic vision in patients with AD [30,34,75]. In 2000, Rizzo et al. [34] measured stereopsis in individuals with mild AD compared to subjects without dementia using the Titmus test. Although a reduction in stereoacuity was observed in the AD group compared to controls, no significant differences were found between groups [34]. Kim et al. [30] assessed the ability of patients with MCI and AD to process coarser disparities than those tested in standard stereopsis tests such as Titmus, TNO, or Randot. In this study, no significant differences were found between the patient group and controls [30]. On the other hand, Lee et al. [75] evaluated stereoscopic vision using a 3D movie and a stereopsis questionnaire and compared the results with the Titmus fly test. While the Titmus test showed no differences between groups, patients with AD scored worse on the stereopsis questionnaire compared to the control group [75]. These findings suggest that traditional methods for evaluating stereopsis might not be sufficiently sensitive to detect functional deficits in AD, possibly due to their limited temporal and spatial resolution or the low cognitive demand involved. However, the use of computer-based tests involving dynamic stereograms, which require greater cortical integration, may offer more sensitive tools for detecting stereoscopic deficits in these patients.

A systematic review published in 2023 reported that visuospatial perception tests may help distinguish between healthy older adults and individuals diagnosed with AD [76]. Notably, the most effective assessment identified was a computerized 3D visual task described by Lemos et al. [77], which showed sensitivity and specificity values of 90% and 95%, respectively.

Other studies have used the Digital Perception Test (DPT) [78] to assess visual integration in patients with AD [36,37]. This test consists of 15 slides, each showing the same image in different spatial positions and distorted thought special effects. The goal of the patient is to identify the image that is correctly oriented. Both studies observed a statistically significant decrease in the test scores among patients with mild and moderate AD compared to the control group. Furthermore, a direct correlation was found between MMSE and DPT scores [36,37]. Analysis using aROC curves also showed good prognostic value, suggesting that the DPT could be considered a reliable test for evaluating visual integration in AD patients. These results may be attributed to disruptions in visual processing in AD, occurring in the parietal and frontal regions, which are mediated by the M pathway [36].

Lastly, it is important to note that patients with AD may also present visual agnosia, which is characterized by difficulty recognizing or identifying objects despite having normal vision. Two main types of visual agnosia can be distinguished: apperceptive and associative. Apperceptive visual agnosia refers to the difficulty in recognizing the shape or structure of an object, while associative visual agnosia involves the inability to link the formal representation of an object to its conceptual meaning. There is some controversy regarding whether visual agnosia in AD is associated with damage to specific cortical circuits or is the result of widespread neuropathological changes [79]. Giannakopoulos et al. [79] observed that associative visual agnosia was related to the formation of NFTs in the occipitotemporal visual association areas, whereas apperceptive visual agnosia appears to be a consequence of diffuse AD pathology. Furthermore, these patients may also have difficulty recognizing familiar faces, a condition known as prosopagnosia [74]. In patients with neurodegenerative diseases, this visual impairment is associated with damage to the temporal lobes [80].

## 9. Limitations and Future Directions

Although visual dysfunction has been observed in AD patients, the current literature has limitations, and further research is needed to better understand the topic.

Currently, AD diagnosis relies on imaging techniques such as positron emission tomography (PET) or cerebrospinal fluid (CSF) biomarker analysis, which are costly and invasive procedures. This underscores the need to explore new diagnostic alternatives, ideally being cost-effective, minimally invasive, and enabling earlier disease detection. Blood biomarkers have advanced significantly in recent years [18]. In this same regard, the eye, due to its close connection to the brain and greater accessibility, has gained attention as a potential source for AD biomarkers.

Specifically, studies have shown that assessing visual function in AD patients could help identify clinical markers useful for the diagnosis of the disease. However, these visual impairments are not specific to AD and may occur in other ocular or systemic conditions. Therefore, it is crucial to strictly select participants for studies evaluating visual function, emphasizing the importance of conducting a prior ophthalmological examination in all patients to rule out any ocular abnormalities that may influence results, thereby avoiding potential bias. Furthermore, due to the lack of specificity in optometric tests, identifying molecular ocular biomarkers, in combination with clinical markers, may offer an accurate and early diagnosis. Recently, tear fluid has emerged as a promising alternative in the search for biomarkers, as it is comparable to CSF. Previous studies have reported that AD patients exhibit alterations in the chemical composition of the tear film (reduced lipocalin-1, lysozyme-C, and lacritin, and elevated dermcidin levels) [81], changes in Aβ and tau protein concentrations [82,83], increased microRNA levels [84], and altered expression of proteins involved in protein repair and clearance system or cytoskeletal regulation [85]. Based on these findings, future research should focus on the study of tear fluid in AD patients to identify biomarkers with high sensitivity and specificity. Moreover, since both visual function and tear fluid can be analyzed using minimally invasive and cost-effective techniques, these tests could be useful for mass screening of at-risk populations, aiming to identify the disease even before the onset of symptoms.

On the other hand, clinical markers of visual function could be useful for monitoring the progression of AD. Specifically, it would be valuable to increase the number of studies evaluating individuals with MCI to identify markers that could predict whether this cognitive impairment may progress to AD or another type of dementia. In this regard, oculomotor control has already shown potential as a method for predicting disease progression [70].

However, one of the main limitations of visual function research in AD is the predominance of cross-sectional studies with small sample sizes, which restrict statistical power and makes it difficult to track disease progression over time. To address this, further research should prioritize longitudinal studies. Specifically, studies that assess oculomotor parameters in at-risk populations, including patients with subjective cognitive decline, could generate more consistent evidence regarding eye movement abnormalities as early biomarkers of cognitive deterioration associated with AD. Moreover, the portability and scalability of modern eye-tracking technologies offer promising opportunities for implementing large-scale, non-invasive assessment protocols.

Another key issue is the use of different techniques to measure optometric parameters in each study. The diversity complicates replication, synthesis, and meta-analytical approaches. Standardizing the protocols employed in these studies is essential to enable more precise and reliable comparisons in future research. It is crucial to select the most appropriate techniques, considering the difficulties of AD patients performing complex tests due to the cognitive limitations associated with the disease.

Lastly, another significant methodological limitation lies in the heterogeneity of diagnostic criteria used across studies. This inconsistency leads to variability in patient characterization and reduced diagnostic accuracy, ultimately limiting the ability to draw reliable conclusions and compare findings across different cohorts.

## 10. Concluding Remarks

The main objective of this review was to gather current knowledge about visual function impairments in AD patients and the potential underlying causes of this visual loss. Overall, it has been observed that AD patients experience a global deterioration of visual function, affecting various optical parameters such as VA, CS, color vision, ocular motility, visual perception, and stereopsis. These dysfunctions are caused by damage to different areas of the visual pathway, from the retina to specific regions of the cerebral cortex.

Given the close relationship between the eye and the brain, ocular studies in neurodegenerative diseases are becoming a promising field in the search for biomarkers for the early identification of these pathologies, as they offer less invasive and cost-effective diagnostic methods. Specifically, the assessment of visual function in AD can contribute to identifying clinical markers, which, together with other molecular and cognitive indicators, would facilitate the diagnosis and prediction of disease progression. Furthermore, the study of visual function also contributes to advancing our understanding of the brain areas affected by the pathology.

Moreover, addressing visual dysfunction in AD is not only relevant for diagnostic purposes but also for improving patient care. Visual impairments can exacerbate cognitive and functional decline, limit patient autonomy, and negatively impact quality of life. Therefore, understanding the visual consequences of AD underscores the need to implement strategies aimed at preserving and enhancing visual function, ultimately helping to prevent further deterioration and support patients’ overall well-being.

Despite the controversy among the reviewed studies, evidence indicates the presence of visual impairment in AD. However, further research is needed, addressing certain limitations, to consolidate these findings. Nevertheless, the observed results highlight the potential of the eye as a window for studying the brain and, specifically, AD.

## Figures and Tables

**Figure 1 jcm-14-05963-f001:**
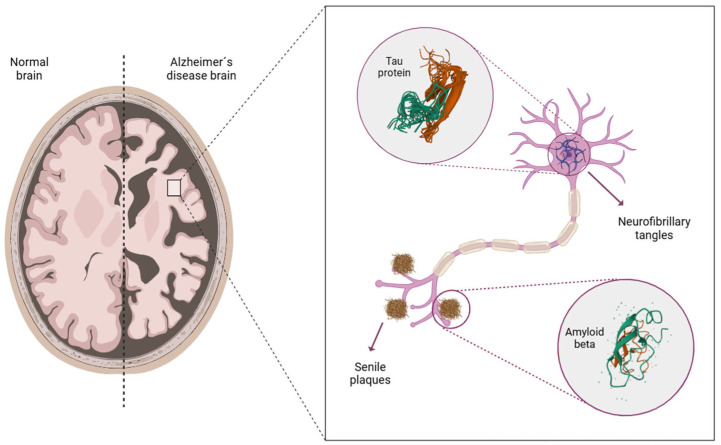
Brain atrophy and formation of senile plaques and neurofibrillary tangles in AD. Figure created by BioRender.com software (Toronto, ON, Canada).

**Figure 2 jcm-14-05963-f002:**
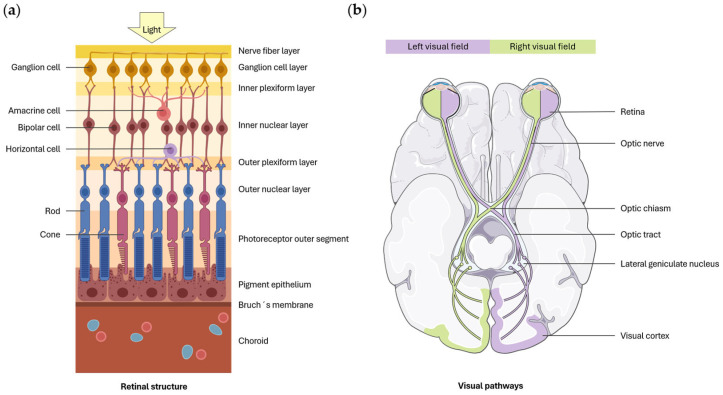
(**a**) Schematic illustration of the retina layers. Figure created by BioRender.com software (Toronto, ON, Canada). (**b**) Schematic representation of visual pathways. From the retina, visual information travels along the optic nerve to the chiasm, where the nasal retinal fibers decussate. It then proceeds through the optic tract to the LGN and, finally, reaches the visual cortex. Self-create image using elements of Servier Medical Art with Creative Commons license.

**Figure 3 jcm-14-05963-f003:**
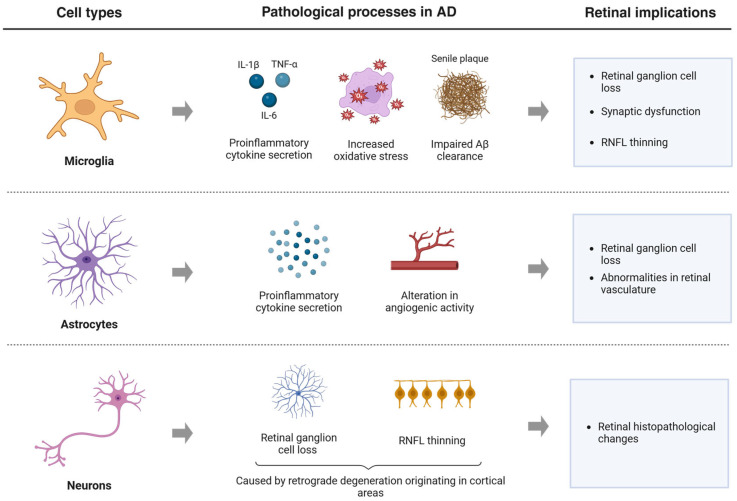
Diagram of the pathological changes in the retina of AD patients mediated by microglia, astrocytes and neurons. Figure created by BioRender.com software (Toronto, ON, Canada).

**Figure 4 jcm-14-05963-f004:**
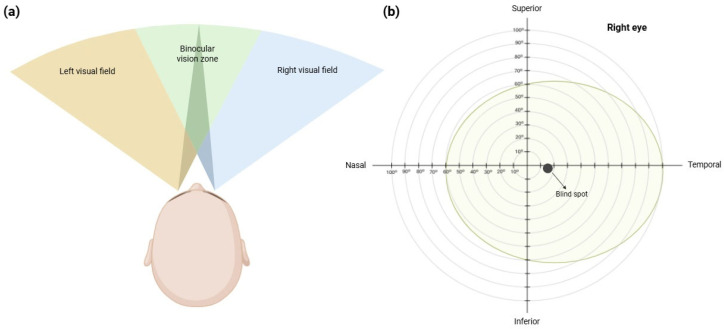
(**a**) Right, left and binocular visual fields. Figure created by BioRender.com software (Toronto, ON, Canada). (**b**) Visual field extension along the superior, inferior, nasal and temporal axes in the right eye. Figure created by BioRender.com software (Toronto, ON, Canada).

**Table 1 jcm-14-05963-t001:** Studies assessing VA in patients with AD.

Study	Sample Size	Test	Main Findings
Rizzo et al. [34]	22 controls 43 mild AD	Sloan letters	No significant differences between groups.
Polo et al. [35]	24 controls 24 mild-moderate AD	LogMAR ETDRS	No significant differences between groups. Correlation between VA and macular and RNFL thickness. No correlation between VA and MMSE.
Salobrar-García et al. [36]	40 controls 39 mild AD 21 moderate AD	Snellen	Significant decrease in VA in patients with mild and moderate AD compared to controls. High predictive value of the VA test. Correlation between VA and MMSE.
Elvira-Hurtado et al. [37]	53 controls 13 FH+ 23 MCI 25 mild AD 21 moderate AD	Snellen	Significant decrease in VA in patients with MCI, mild AD and moderate AD compared to controls. No significant differences between FH+ group and controls. High predictive value of the AV test. Correlation between VA and MMSE.
Nolan et al. [38]	33 controls 36 moderate AD	LogMAR ETDRS	Significant decrease in VA in patients with moderate AD compared to controls

AD: Alzheimer’s disease; FH+: familiar history positive; MMSE: Mini-Mental State Examination; RNFL: retinal nerve fiber layer; VA: visual acuity.

**Table 2 jcm-14-05963-t002:** Studies assessing CS in patients with AD.

Study	Sample Size	Test	Main Findings
Polo et al. [35]	24 controls 24 mild-moderate AD	Pelli-Robson CSV-1000E	Significant reduction in CS in AD patients with Pelli-Robson test. Significant reduction in CS in AD patients at lower spatial frequencies (3, 6, 12 cpd) with CSV-1000E test. Correlation between CS and macular and RNFL thickness. No correlation between CS and MMSE.
Salobrar-García et al. [36]	40 controls 39 mild AD 21 moderate AD	CSV-1000E	Significant decrease in CS in patients with mild and moderate AD at all the spatial frequencies tested (3, 6, 12, 18 cpd). High predictive value of CS test, especially at high spatial frequencies. Correlation between CS and MMSE.
Elvira-Hurtado et al. [37]	53 controls 13 FH+ 23 MCI 25 mild AD 21 moderate AD	CSV-1000E	Significant differences in higher spatial frequencies (6, 12, 18 cpd) of CS between the controls and AD groups. Differences between MCI and controls only at 12 and 18 cpd. High predictive value of CS test, especially at high spatial frequencies. Correlation between CS and MMSE.
Nolan et al. [38]	33 controls 36 moderate AD	LogMAR ETDRS	Significant decrease in CS in patients with moderate AD at all spatial frequencies.
Hutton et al. [43]	6 controls 6 mild-moderate AD	Computerized contrast sensitivity system	Significant decrease in CS in patients with mild-moderate AD at high spatial frequencies.
Schlotterer et al. [44]	11 young controls 11 elderly controls 10 AD	-	No significant differences between the elderly controls and AD patients.

AD: Alzheimer’s disease; CS: contrast sensitivity; MMSE: Mini-Mental State Examination; RNFL: retinal nerve fiber layer.

**Table 3 jcm-14-05963-t003:** Studies assessing color vision in patients with AD.

Study	Sample Size	Test	Main Findings
Polo et al. [35]	24 controls 24 mild-moderate AD	Color Vision Recorder (Farnsworth and L’Anthony)	Significant impairment of color vision in patients with AD. Correlation between color vision and RNFL and macular parameters in isolated sectors. No correlation between color vision and MMSE.
Salobrar-García et al. [36]	40 controls 39 mild AD 21 moderate AD	Roth 28-Hue	Higher number of total nonspecific, tritan axis and deutan axis errors in patients with mild and moderate AD. High predictive value of color vision test. Correlation between color vision test errors and MMSE.
Elvira-Hurtado et al. [37]	53 controls 13 FH+ 23 MCI 25 mild AD 21 moderate AD	Farnsworth 28-Hue	Higher number of total nonspecific, tritan axis and deutan axis errors in AD patients. No significant differences between FH+ group and controls. High predictive value of the color vision test. Correlation between color vision test errors and MMSE.
Pache et al. [49]	25 controls 26 mild-severe AD	Ishihara PV-16	Higher number of total nonspecific errors in AD patients. No correlation between color vision and severity of the disease.
Bassi et al. [50]	10 young controls 11 age-matched controls 10 probable AD 10 other dementias	L’Anthony D-15	No significant differences between patients with AD and age-matched controls.
Wijk et al. [51]	12 controls 12 AD	NCS color order system	No significant differences between groups.
Salobrar-García et al. [52]	28 controls 23 mild-AD	Roth 28-Hue	Higher number of tritan nonspecific errors in AD patients. High predictive value of the tritan errors. Correlation between tritan errors and MMSE.
Vidal et al. [53]	18 controls 23 MCI 13 AD	Cambridge Color Test	Poorer color vision in AD patients along the protan, deutan, and tritan axes compared to controls.

AD: Alzheimer’s disease; FH+: familiar history positive; MMSE: Mini-Mental State Examination; RNFL: retina nerve fiber layer.

**Table 4 jcm-14-05963-t004:** Studies assessing ocular motility in patients with AD.

Study	Sample Size	Test	Main Findings
Qi et al. [61]	42 controls 63 MCI due to AD 49 dementia due to AD	EyeKnow	Abnormalities in lateral fixation, prosaccades, antisaccades, and memory saccades in AD patients. Correlation between eye movement alterations and global cognition, various cognitive domains, and daily living activities. High predictive value of the eye movement alterations.
Tao et al. [62]	20 controls 23 MCI 23 AD	EyeKnow	Alterations in lateral fixation, prosaccades, and antisaccades in AD patients. No differences in smooth pursuit movements. Correlation between some oculomotor parameters and cognitive function. High predictive value of the combination of lateral fixation and antisaccades.
Zuben et al. [63]	26 controls 47 MCI 18 AD	Tobii TX300	Differences in eye movement behavior between control subjects and patients with MCI and AD.
Parra et al. [65]	42 controls 65 MCI	ViewMind	Differences in fixation and saccades between groups. Oculomotor behavior analysis can effectively predict the progression to AD.
Hannonen et al. [66]	37 controls 20 MCI 21 mild AD	Tobii TX300	Differences in saccades between control subjects and patients with MCI and AD.
Ma et al. [67]	34 controls 80 mil-moderate AD	EyeKnow	Abnormalities in prosaccades and antisaccades in AD patients. No differences in pursuit and fixation tasks. Prosaccades and antisaccades were correlated with global cognitive function and specific cognitive domains.

AD: Alzheimer’s disease; MCI: mild cognitive impairment.

## Data Availability

Not applicable.

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
