# Peer review of "Visual Function in Alzheimer’s Disease: Current Understanding and Potential Mechanisms Behind Visual Impairment"

_jcm, 2025, doi:10.3390/jcm14175963_

Round 1
Reviewer 1 Report
Comments and Suggestions for Authors
Is this article a narrative review? The manuscript could be best classified as a narrative review. It does not follow the format or rigor typically associated with systematic reviews (e.g., PRISMA guidelines, structured search strategy, inclusion/exclusion criteria), nor does it present new experimental or observational data. The article instead provides a broad overview of current knowledge on a specific clinical topic, drawing from selected literature to support discussion.
If accepted as a narrative review, the article must ensure that the synthesis is critical and not merely descriptive, and the authors should make that distinction clear in the title, abstract, and methods section.
In general, the manuscript addresses an important area within ophthalmology, specifically focusing on the potential of visual function assessment as a tool for diagnosis or predicting Alzheimer´s disease progression. The topic is timely and relevant to both clinicians and researchers. The review is informative, but requires substantive improvements to meet the journal’s standards for peer-reviewed narrative reviews.
Major Comments
- Clarification of Article Type and Methods
- The manuscript lacks a clear definition of its review type. The authors should state explicitly that this is a narrative review and briefly describe their literature search approach (even if not systematic).
- Add the aim of this review to the abstract and at the end of the Introduction.
- Please add methods, results, and discussion sections, according to the narrative review guidelines.
- Methods: There is no mention of databases searched, inclusion criteria, or time frame of literature considered. While this is acceptable for a narrative review, transparency is still expected. These
- Results: Parts 2 to 9 should become subparts of the results.
- Depth of Critical Analysis
- Discussion: The article tends to summarize existing studies rather than critically evaluate them. A more analytical discussion of the strengths, limitations, and clinical applicability of the cited studies is needed.
Generally clear, but minor grammatical and syntactic issues are present. A professional language edit is recommended.
Author Response
The authors would like to thank the reviewer for the comments and highlights to strengthen our manuscript and are grateful for the time spent reviewing this article. The reviewer´s comments have been addressed individually below.
Comment 1: The manuscript lacks a clear definition of its review type. The authors should state explicitly that this is a narrative review and briefly describe their literature search approach (even if not systematic).
We appreciate the reviewer´s comment. We confirm that this manuscript is a narrative review, and we have now explicitly indicated this in the Abstract to avoid any ambiguity. Line 28 (Manuscript marked copy).
Regarding the literature search approach, we acknowledge the reviewer´s suggestion and fully understand its importance. In this case, we followed the journal´s author guidelines, which indicate that narrative reviews are not required to include a methods section unless the review is systematic. Based on this, and to maintain a concise and focused structure aligned with the journal´s format, we did not include a detailed description of the search strategy in the manuscript.
We hope this explanation clarifies our approach, and we remain open to further adjustments if the editorial team considers it necessary.
Comment 2: Add the aim of this review to the abstract and at the end of the Introduction.
We thank the reviewer for this helpful suggestion. In response, we have stated the aim of the review in both the Abstract (lines 28-30 and 35-39, Manuscript marked copy) and the final paragraph of the Introduction (lines 138-147 (Manuscript marked copy)), to enhance the reader´s understanding of the article´s purpose and scope.
Comment 3: Please add methods, results, and discussion sections, according to the narrative review guidelines.
We value the reviewer´s recommendation. After consulting the Journal of Clinical Medicine author guidelines, we understand that narrative reviews are not required to follow a fixed structure. The journal indicates that the review should be organized logically, without the obligation to include specific sections, such as Methods or Results, unless it is a systematic review.
Regarding the Discussion section, the guidelines do mention that it can be included (The structure can include an Abstract, Keywords, Introduction, Relevant Sections, Discussion, Conclusions, and Future Directions.); however, it is not presented as mandatory. In our manuscript, the discussion has been integrated within the section “Limitations and Future Directions”.
Given the narrative nature of our manuscript, we chose a structure that we believe best supports a coherent and thematic presentation of the topic. Nonetheless, we remain fully open to reformatting the manuscript into more defined sections if the editorial board considers it necessary for consistency or clarity.
Comment 4 & 5: Methods: There is no mention of databases searched, inclusion criteria, or time frame of literature considered. While this is acceptable for a narrative review, transparency is still expected. Results: Parts 2 to 9 should become subparts of the results.
We thank the reviewer for highlighting the importance of transparency in narrative reviews. As previously explained, and in line with the Journal of Clinical Medicine guidelines, specific methodological details are not mandatory for this type of review and therefore were not included.
Regarding the organization of sections 2 to 9, we structured the manuscript to present the content in a clear and coherent manner appropriate for a narrative review. Nevertheless, we remain willing to adjust the structure to better align with the journal´s preferences if deemed necessary by the editorial team.
Comment 6: Discussion: The article tends to summarize existing studies rather than critically evaluate them. A more analytical discussion of the strengths, limitations, and clinical applicability of the cited studies is needed.
We thank the reviewer for this important observation. In response, we have made substantial modifications throughout the manuscript to provide a more critical perspective on the studies evaluated. All changes and corrections throughout the body of the manuscript are marked and highlighted in the document entitled “Manuscript marked”. Specifically, regarding the Discussion, this is incorporated within the section titled “Limitations and Future Directions”, where we have added two new paragraphs to enhance the analytical depth: one between lines 714-722 and another between lines 730-733 (Manuscript marked copy).
Comment 7: Comments on the Quality of English Language. Generally clear, but minor grammatical and syntactic issues are present. A professional language edit is recommended.
The entire manuscript has been carefully reviewed, and some grammatical and syntactic corrections have been made to improve the readability. All changes are marked in the document entitled “Manuscript marked”.
Reviewer 2 Report
Comments and Suggestions for Authors
The manuscript by Alvite-Piñeiro et al. compiles a comprehensive overview of currently known visual function changes in Alzheimer’s disease (AD), spanning from retinal alterations to cortical involvement. It describes a range of articles across multiple visual domains, including acuity, contrast sensitivity, colour vision, visual fields, oculomotor function, and perception. The authors contextualised these findings within the broader neuroanatomical and pathophysiological framework of AD. The section on oculomotor control is particularly well-developed, integrating data from multiple eye-tracking technologies and including quantitative metrics, which help to characterise the predictive utility of these measures in early-stage MCI and AD. The manuscript is well written and is easy to read, like a chapter in a general knowledge book.
The stated aim is to provide a synthesis of current knowledge regarding visual function in AD but I do not find a novel component in it. The manuscript title promises a review of potential mechanisms, though these are not described. I think that the section reviewing from retina to cortex, remains very basic and reiterates findings that are already well-known in the field. For example, figures 1 to 5 are standard schematic illustrations that do not add new insight or meaningfully support the scientific narrative related to AD. Replacing the current figures with novel presentation of data, such as diagrams illustrating specific visual pathways or cellular mechanisms affected in AD would make the figures more relevant to the topic reviewed.
Key sections, particularly those on visual information processing, visual acuity, contrast sensitivity, and colour vision largely summarise existing literature without offering new interpretations, or critical comparisons between studies, or meta-analytical synthesis, or evaluation of methodological variability or quality. Many of the findings cited such as MMSE correlations, tritan axis deficits, or retinal ganglion cell thinning have been extensively documented in previous reviews dating back to 2019. More literature on these visual problems in AD has emerged, reinforcing that visual problems in AD exist, but they are not described here with the novelty they should. Without advancing these discussions through critical appraisal, or the introduction of underexplored perspectives, the manuscript does not provide the level of originality expected in an updated review of this topic.
A more impactful contribution could be achieved by narrowing the focus to a less-reviewed area (oculomotor biomarkers), enhancing critical comparisons between studies, incorporating summary visual models, and framing the review around a central potential mechanism question.
Author Response
The authors are grateful for the reviewer´s interest in our article and the constructive feedback. Many thanks also for the careful review that contributes to improving our manuscript.
Comment 1: The stated aim is to provide a synthesis of current knowledge regarding visual function in AD but I do not find a novel component in it. The manuscript title promises a review of potential mechanisms, though these are not described.
We appreciate the reviewer´s observation. To address this concern, we have thoroughly revised the manuscript to incorporate updated content and to offer a more critical and integrative perspective. Across the different sections, we have expanded the discussion of study limitations and conducted a more in-depth comparison of findings, aiming to provide a clearer and more original synthesis of the current evidence (Manuscript marked).
Regarding the potential mechanisms, we have aimed to incorporate information throughout the different sections of the manuscript about potential retinal and cortical changes that may contribute to visual dysfunction in Alzheimer´s disease, in line with the focus suggested in the title. Specifically, new content addressing this point has been added to the sections on “Visual acuity” (lines 328-256), “Contrast sensitivity” (lines 361-364), and “Visual field” (lines 532-547) (Manuscript marked copy).
Comment 2: I think that the section reviewing from retina to cortex remains very basic and reiterates findings that are already well-known in the field. For example, figures 1 to 5 are standard schematic illustrations that do not add new insight or meaningfully support the scientific narrative related to AD. Replacing the current figures with novel presentation of data, such as diagrams illustrating specific visual pathways or cellular mechanisms affected in AD would make the figures more relevant to the topic reviewed.
We agree with the reviewer´s comment that the section “Visual Information Processing” presents basic and well-known information. However, we believe that including this overview is important to provide readers, especially those less familiar with the topic, with essential background and context for understanding the subsequent, more detailed discussions.
Regarding Figures 1 to 5, we have taken the reviewer´s suggestion into account and decided to remove Figures 3 and 4, as they provided relatively simple illustrations. Additionally, we have included a new figure in the section “Visual acuity” (now Figure 3), which outlines pathological mechanisms affecting the retina in Alzheimer´s disease patients.
Comment 3: Key sections, particularly those on visual information processing, visual acuity, contrast sensitivity, and colour vision largely summarise existing literature without offering new interpretations, or critical comparisons between studies, or meta-analytical synthesis, or evaluation of methodological variability or quality.
We acknowledge the reviewer´s point and have made the necessary revisions to strengthen the manuscript. In response, we have revised the sections “Visual acuity” (lines 228-236), “Contrast sensitivity” (lines 314-336), and “Color vision” (lines 419-425, lines 429-442) to incorporate a more critical analysis of the cited studies (Manuscript marked). These revisions include a discussion of study´s limitations, the methodological variability that hinders direct comparisons between findings, and an evaluation of the overall quality of the methods used to assess visual function in cognitively impaired populations.
Comment 4: Many of the findings cited such as MMSE correlations, tritan axis deficits, or retinal ganglion cell thinning have been extensively documented in previous reviews dating back to 2019.
We recognize that several of the findings referenced, such as MMSE correlations, tritan axis deficits, and thinning of the retinal nerve fiber layer, have been consistently reported in earlier literature. Nevertheless, we consider their inclusion essential to provide a comprehensive and contextually grounded review that integrates foundational and recent findings. These elements establish the necessary background for readers and support the critical insights that follow. Importantly, we have aimed to go beyond prior reviews by incorporating updated data, comparing results across studies, and offering a more nuanced and critical perspective that reflects the current state of knowledge in the field.
Comment 5: More literature on these visual problems in AD has emerged, reinforcing that visual problems in AD exist, but they are not described here with the novelty they should. Without advancing these discussions through critical appraisal, or the introduction of underexplored perspectives, the manuscript does not provide the level of originality expected in an updated review of this topic. A more impactful contribution could be achieved by narrowing the focus to a less-reviewed area (oculomotor biomarkers), enhancing critical comparisons between studies, incorporating summary visual models, and framing the review around a central potential mechanism question.
We thank the reviewer for this insightful comment. As mentioned previously, we have carried out a comprehensive revision of the manuscript, incorporating substantial changes aimed at enhancing its critical depth and originality. These modifications include updated content, and a more analytical evaluation of the cited studies. All changes are clearly highlighted in the “Manuscript marked” document for ease of review.
Reviewer 3 Report
Comments and Suggestions for Authors
This revised manuscript is a very successful and interesting summary of clinical value concerning possible changes in vision in patients with AD, such as changes in visual acuity, contrast sensitivity, color vision, visual field, ocular mobility, visual perception, and stereopsis. Limitations are mentioned. Patients with AD experience a global deterioration of visual functions, caused by damage to different areas of the visual pathway, from the retina to specific regions of the cerebral cortex. The assessment of visual function in AD can contribute to identifying clinical biomarkers.
Comments:
Line 151: … stimulate “via the bipolar cells” the retinal ganglion cells….
Typos: Line 265: the the; 529: bio-marker; 636: specificity; 660: prosop-agnosia; 680: markers; 715:dia-gnostic
Why were Fig.3. and 4. from Version rev1 removed? They would be informative…
Author Response
The authors thank the reviewer for their interest in our article and for their helpful comments. The suggestions have been addressed to improve the clarity and accuracy of the manuscript.
Comment 1: Line 151: … stimulate “via the bipolar cells” the retinal ganglion cells….
We thank the reviewer for their comment. This information has been revised and expanded as follows: “These electrical impulses stimulate the retinal ganglion cells (RGCs) via the bipolar cells, while horizontal and amacrine cells modulate this transmission.” (lines 150-152, Manuscript marked copy). All changes are clearly highlighted in blue in the “Manuscript marked” document.
- Typos: Line 265: the the; 529: bio-marker; 636: specificity; 660: prosop-agnosia; 680: markers; 715: dia-gnostic
We thank the reviewer for pointing out these typographical errors. All suggested corrections have been made and are highlighted in blue in the “Manuscript marked” document, except for lines 529, 660, and 715. In these cases, the hyphens appear automatically due to the formatting of the document template provided by the Journal of Clinical Medicine; therefore, these changes cannot be modified.
- Why were Fig.3. and 4. from Version rev1 removed? They would be informative…
We appreciate the reviewer´s observation. Figures 3 and 4 from Version 1 of the manuscript were removed in response to the comments for the Reviewer 2, who noted: “For example, figures 1 to 5 are standard schematic illustrations that do not add new insight or meaningfully support the scientific narrative related to AD. Replacing the current figures with novel presentation of data, such as diagrams illustrating specific visual pathways or cellular mechanisms affected in AD would make the figures more relevant to the topic reviewed.”
Accordingly, we decided to remove Figures 3 and 4, as they provided relatively simple illustrations. In addition, we have included the Figure 3 in the section “Visual acuity”, which outlines pathological mechanisms affecting the retina in Alzheimer´s disease patients.
Round 2
Reviewer 1 Report
Comments and Suggestions for Authors
All comments have been addressed. Since it was clarified that the manuscript is a narrative and not systematic review, the structure is accepted. This manuscript could be accepted in the present version.
Author Response
The authors sincerely thank the reviewer for their constructive comments, for acknowledging that the manuscript is suitable for publication in its current form, and for the time spent reviewing this article.